# PuLID: Pure and Lightning ID Customization via Contrastive Alignment

**Zinan Guo**[*]   **Yanze Wu**[*†]   **Zhuowei Chen**   **Lang Chen**   **Peng Zhang**   **Qian He**
ByteDance Inc.
guozinan.1@bytedance.com   wuyanze123@gmail.com

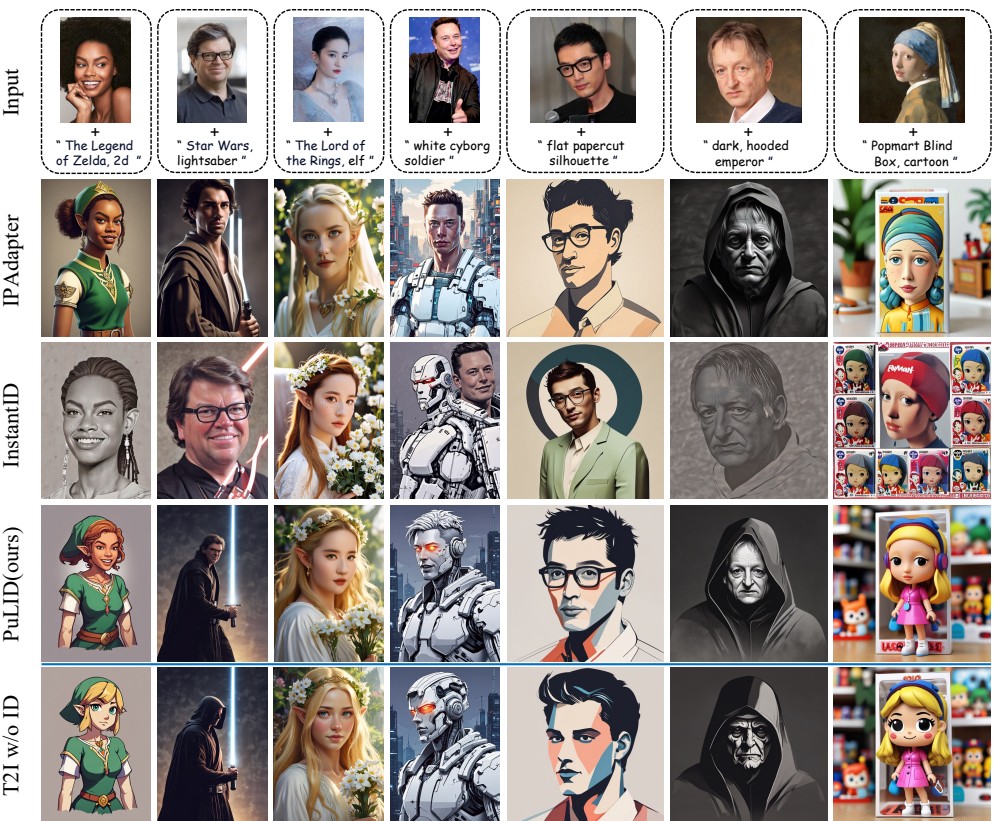

Figure 1: We introduce PuLID, a tuning-free ID customization approach. PuLID maintains high ID fidelity while effectively reducing interference with the original model's behavior.

## Abstract

We propose Pure and Lightning ID customization (PuLID), a novel tuning-free ID customization method for text-to-image generation. By incorporating a Lightning T2I branch with a standard diffusion one, PuLID introduces both contrastive alignment loss and accurate ID loss, minimizing disruption to the original model and ensuring high ID fidelity. Experiments show that PuLID achieves superior performance in both ID fidelity and editability. Another attractive property of PuLID is that the image elements (*e.g.*, background, lighting, composition, and style) before and after the ID insertion are kept as consistent as possible. Codes and models are available at https://github.com/ToTheBeginning/PuLID.

---

[*]Equal contributions. [†] Corresponding author

38th Conference on Neural Information Processing Systems (NeurIPS 2024).

# 1 Introduction

As a special category of customized text-to-image (T2I) generation [7, 36, 14, 19, 46, 50], identity (ID) customization allow users to adapt pre-trained T2I diffusion models to align with their personalized ID. One line of work [7, 36, 14, 19] fine-tunes certain parameters on several images with the same ID provided by the user, thereby embedding the ID into the generative model. These methods have spawned many popular AI portrait applications, such as PhotoAI and EPIK.

While tuning-based solutions have achieved commendable results, customizing for each ID requires tens of minutes of fine-tuning, thus making the personalization process economically expensive. Another line of work [48, 50, 3, 42, 22, 21, 44] forgoes the necessity of fine-tuning for each ID, instead resorting to pre-training an ID adapter [13, 28] on an expansive portrait dataset. These methods typically utilize an encoder (*e.g.*, CLIP image encoder [32]) to extract the ID feature. The extracted feature is then integrated into the base diffusion model in a specific way (*e.g.*, embedded into cross-attention layer). Although highly efficient, these tuning-free methods face two significant challenges.

• **Insertion of ID disrupts the original model's behavior**. A pure ID information embedding should feature two characteristics. Firstly, an ideal ID insertion should alter only ID-related aspects, such as face, hairstyle, and skin color, while image elements not directly associated with the specific identity, such as background, lighting, composition, and style, should be consistent with the behavior of the original model. To our knowledge, this point has not been focused by previous works. While some research [50, 44, 22] has shown the ability for stylized ID generation, notable style degradation occurs when compared with images before ID insertion (as depicted in Fig. 1). Methods with higher ID fidelity tend to induce more severe style degradation.

Secondly, after the ID insertion, it should still retain the ability of the original T2I model to follow prompts. In the context of ID customization, this generally implies the capacity to alter ID attributes (*e.g.*, age, gender, expression, and hair), orientation, and accessories (*e.g.*, glasses) via prompts. To achieve these features, current solutions generally fall into two categories. The first category involves enhancing the encoder. IPAdapter [50, 1] shifted from early-version CLIP extraction of grid features to utilizing face recognition backbone [6] to extract more abstract and relevant ID information. Despite the improved editability, the ID fidelity is not high enough. InstantID [44] builds on this by including an additional ID&Landmark ControlNet [52] for more effective modulation. Even though the ID similarity improves significantly, it compromises some degree of editability and flexibility. The second category of methods [22] supports non-reconstructive training to enhance editability by constructing datasets grouped by ID; each ID includes several images. However, creating such datasets demands significant effort. Also, most IDs correspond to a limited number of celebrities, which might limit their effectiveness on non-celebrities.

• **Lack of ID fidelity**. Given our human sensitivity to faces, maintaining a high degree of ID fidelity is crucial in ID customization tasks. Inspired by the successful experience of face generation [35, 45] tasks during the GAN era [9], a straightforward idea for improving ID fidelity is to introduce ID loss within diffusion training. However, due to the iterative denoising nature of diffusion models [12], achieving an accurate $x_0$ needs multiple steps. The resource consumption for training in this manner can be prohibitively high. Consequently, some methods [3] predict $x_0$ directly from the current timestep and then calculate the ID loss. However, when the current timestep is large, the predicted $x_0$ is often noisy and flawed. Calculating ID loss under such conditions is obviously inaccurate, as the face recognition backbone [6] is trained on photo-realistic images. Although some workarounds have been proposed, such as calculating ID loss only at less noisy timesteps [29] or predicting $x_0$ with an additional inference step [54], there still remains room for improvement.

In this work, to maintain high ID fidelity while reducing the influence on the original model's behavior, we propose **PuLID**, a pure and lighting ID customization method via contrastive alignment. Specifically, we introduce a **Lightning T2I branch** alongside the standard diffusion-denoising training branch. Leveraging recent fast sampling methods [26, 38, 23], the lighting T2I branch can generate high-quality images from pure noise with a limited and manageable number of steps. With this additional branch, we can simultaneously address the two challenges mentioned above. Firstly, to minimize the influence on the original model's behavior, we construct a contrastive pair with the same prompt and initial latent, with and without ID insertion. During the Lightning T2I process, we align the UNet features between the contrastive pair semantically, instructing the ID adapter how to insert

ID information without affecting the behavior of the original model. Secondly, as we now have the precise and high-quality generated $x_0$ after ID insertion, we can naturally extract its face embedding and calculate an accurate ID loss with the ground truth face embedding. It is worth mentioning that such $x_0$ generation process aligns with the actual test setting. Our experiments demonstrate that optimizing the ID loss in this context can significantly increase ID similarity.

The contributions are summarized as follows. **(1)** We propose a tuning-free method, namely, PuLID, which preserves high ID similarity while mitigating the impact on the original model's behavior. **(2)** We introduce a Lightning T2I branch alongside the regular diffusion branch. Within this branch, we incorporate a contrastive alignment loss and ID loss to minimize the contamination of ID information on the original model while ensuring fidelity. Compared to the current mainstream approaches that improve the ID encoder or datasets, **we offer a new perspective and training paradigm**. **(3)** Experiments show that our method achieves SOTA performance in terms of both ID fidelity and editability. Moreover, compared to existing methods, our ID information is less invasive to the model, making our method more flexible for practical applications.

## 2 Related Work

**Tuning-free Text-to-image ID Customization.** ID Customization for text-to-image models aims to empower pre-trained models to generate images of specific identities while following the text descriptions. To ease the resource demand necessitated by tuning-based methods [7, 36, 14, 19, 10, 41], a series of tuning-free methods [42, 44, 29, 50, 22, 48, 51, 4] have emerged, which directly encode ID information into the generation process. The major challenge these methods encounter is minimizing disruption to the original behavior of T2I models while still maintaining high ID fidelity.

In terms of minimizing the disruption, one plausible approach is to utilize a face recognition model [6] to extract more abstract and relevant facial domain-specific representations, as done by IP-Apdater-FaceID [1] and InstantID [44]. A dataset comprising multiple images from the same ID can facilitate the learning of a common representation [22]. Despite the progress made by these approaches, they have yet to fundamentally solve the disruption issue. Notably, models with higher ID fidelity often cause more significant disruptions to the behavior of the original model. In this study, we propose a new perspective and training method to tackle this issue. Interestingly, the suggested method does not require laborious dataset collection grouped by ID, nor is it confined to a specific ID encoder.

To improve ID fidelity, ID loss is employed in previous works [18, 3], motivated by its effectiveness in prior GAN-based works [35, 45]. However, in these methods, $x_0$ is typically directly predicted from the current timestep using a single step, often resulting in noisy and flawed images. Such images are not ideal for the face recognition models [6], as they are trained on real-world images. PortraitBooth [29] alleviates this issue by only applying ID loss at less noisy stages, which ignores such loss in the early steps, thereby limiting its overall effectiveness. Diffswap [54] obtains a better predicted $x_0$ by employing two steps instead of just one, even though this estimation still contains noisy artifacts. In our work, with the introduced Lightning T2I training branch, we can calculate ID loss in a more accurate setting.

We notice a concurrent work, LCM-Lookahead [8], which also uses fast sampling technology (*i.e.*, LCM [26]) to achieve a more precise prediction of $x_0$. However, there are several differences between this work and ours. Firstly, LCM-Lookahead makes a precise prediction of $x_0$ during the conventional diffusion-denoising process, whereas we start from pure noise and iteratively denoise to $x_0$. Our approach, which aligns better with actual testing settings, makes the optimization of ID loss more direct. Secondly, to enhance prompt editing capability, LCM-Lookahead capitalized on the mode collapse phenomenon of SDXL-Turbo [38] to synthesis an ID-consistent dataset. However, the synthetic dataset might face diversity and consistency challenges, and the authors found that training with this dataset may lean towards stylized results more frequently than other methods. In contrast, our method does not need an ID-grouped dataset. Instead, we enhance prompt follow ability through a more fundamental and intuitive approach, namely, contrastive alignment.

**Fast Sampling of Diffusion Models.** In practice, diffusion models are typically trained under 1000 steps. During inference, such a lengthy process can be shortened to a few dozen steps with the help of advanced sampling methods [39, 25, 17]. Recent distill-based works [23, 26, 38] further accelerate this generation process within 10 steps. The core motivation is to guide the student network to align with points further from the base teacher model. In this study, the Lightning T2I training branch we

introduce leverages the SDXL-Lightning [23] acceleration technology, thus enabling us to generate high-quality images from pure noise in just $4$ steps.

# 3 Methods

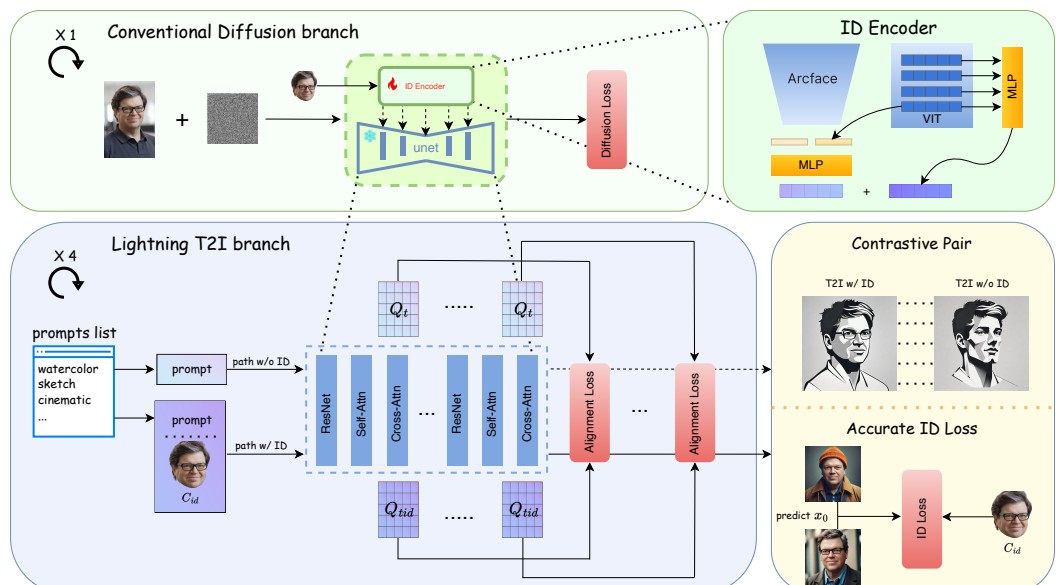

Figure 2: **Overview of PuLID framework.** The upper half of the framework illustrates the conventional diffusion training process. The face extracted from the same image is employed as the ID condition $C_{id}$. The lower half of the framework demonstrates the Lightning T2I training branch introduced in this study. It leverages the recent fast sampling methods to iteratively denoise from pure noise to high-quality images in a few steps (4 in this paper). In this branch, we construct contrastive paths with and without ID injection and introduce an alignment loss to instruct the model on how to insert ID condition without disrupting the original model's behavior. As this branch can produce photo-realistic images, it implies that we can achieve a more accurate ID loss for optimization.

## 3.1 Preliminary

**Diffusion models** [12] are a class of generative models capable of synthesizing desired data samples through iterative denoising. A conventional diffusion training encapsulates two procedures, the forward diffusion process and reverse denoising process. During the diffusion process, noise $\epsilon$ is sampled and added to the data sample $x_0$ based on a predefined noise schedule. This process yields a noisy sample $x_t$ at timestep $t$. Conversely, during the denoising process, a denoisng model $\epsilon_\theta$ takes $x_t$, $t$, and optional additional conditions $C$ as inputs to predict the added noise, the optimization process can be articulated as:

$$\mathcal{L}_{\text{diff}} = \mathrm{E}_{x_0,\epsilon,t}(\|\epsilon - \epsilon_\theta(x_t, t, C)\|). \tag{1}$$

The denoising model $\epsilon_\theta$ in modern T2I diffusion models [37, 33, 30] is predominantly a UNET composed of residual blocks [11], self-attention layers, and cross-attention [43] layers. The prompt, as a condition, is embedded into the cross-attention layers adhering to the attention mechanism, illustrated as follows:

$$\begin{cases} \text{Attention}(Q, K, V) = \text{Softmax}(\frac{QK^T}{\sqrt{d}})V \\ K = \mathbf{W}_K \tau_{txt}(C_{txt}); \ V = \mathbf{W}_V \tau_{txt}(C_{txt}), \end{cases} \tag{2}$$

where $Q$ is projected from the UNET image features, $\tau_{txt}$ denotes a pre-trained language model that converts prompt $C_{txt}$ to textual features, $\mathbf{W}_K$ and $\mathbf{W}_V$ are the learned linear layers.

**ID Customization** in T2I diffusion introduces ID images $C_{id}$ as an additional condition, working together with the prompt to control image generation. Tuning-free customization [16, 48, 50] methods

typically employ an encoder to extract ID features from $C_{id}$. The encoder often includes a frozen backbone, such as CLIP image encoder [32] or face recognition backbone [6], along with a learnable head. A simple yet effective technique to embed the ID features to the pre-trained T2I model is to add parallel cross-attention layers to the original ones. In these parallel layers, learnable linear layers are introduced to project the ID features into $K_{id}$ and $V_{id}$ for calculating attention with $Q$. This technique, proposed by IP-Adapter [50], has been widely used, we also adopt it for embedding ID features in this study.

## 3.2 Basic Settings

We build our model based on the pre-trained SDXL [30], which is a SOTA T2I latent diffusion model. Our ID encoder employs two commonly used backbones within the ID customization domain: the face recognition model [6] and the CLIP image encoder [32], to extract ID features. Specifically, we concatenate the feature vectors from the last layer of both backbones (for the CLIP image encoder, we use the CLS token feature), and employ a Multilayer Perceptron (MLP) to map them into $5$ tokens as the global ID features. Additionally, following ELITE's approach [46], we use MLPs to map the multi-layer features of CLIP to another $5$ tokens, serving as the local ID features. It is worth noting that our method is not restricted to a specific encoder.

## 3.3 Discussion on Common Diffusion Training in ID Customization

Currently, tuning-free ID customization methods generally face a challenge: the embedding of the ID disrupts the behavior of the original model. The disruption manifests in two ways: firstly, the ID-irrelevant elements in the generated image (e.g., background, lighting, composition, and style) have changed extensively compared to before the ID insertion; secondly, there is a loss of prompt adherence, implying we can hardly edit the ID attributes, orientations, and accessories with the prompt. Typically, models with higher ID fidelity suffer more severe disruptions. Before we present our solutions, we first analyze why conventional diffusion training would cause this issue.

In conventional ID Customization diffusion training process, as formulated in Eq. 1, the ID condition $C_{id}$ is usually cropped from the target image $x_0$ [50, 44]. In this scenario, the ID condition aligns completely with the prompt and UNET features, implying the ID condition does not constitute contamination to the T2I diffusion model during the training process. This essentially forms a reconstruction training task. So, to better reconstruct $x_0$ (or predict noise $\epsilon$), the model will make the utmost effort to use all the information from ID features (which may likely contain ID-irrelevant information), as well as bias the training parameters towards the dataset distribution, typically in the realistic portrait domain. Consequently, during testing, when we provide a prompt that is in conflict or misaligned with the ID condition, such as altering ID attributes or changing styles, these methods tend to fail. This is because there exists a disparity between the testing and training settings.

## 3.4 Uncontaminated ID Insertion via Contrastive Alignment

While it is difficult to ascertain whether the insertion of ID disrupts the original model's behavior during the conventional diffusion training, it is rather easy to recognize under the test settings. For instance, we can easily observe whether the elements of the image change after the ID is embedded, and whether it still possesses prompt follow ability. Thus, our solution is intuitive. We introduce a Lightning T2I training branch beyond the conventional diffusion-denoising training branch. Just like in the test setting, the Lighting T2I branch starts from pure noise and goes through the full iterative denoising steps until reaching $x_0$. Leveraging recent fast sampling methods [26, 38, 23], the Lighting T2I branch can generate high-quality images from pure noise with a limited and manageable number of steps. Concretely, we employ SDXL-Lightning [23] with $4$ denoising steps. We prepare a list of challenging prompts that can easily reveal contamination, as shown in Table 8. During each training iteration, a random prompt from this list is chosen as the textual condition for the Lightning T2I branch. Then, we construct contrastive paths that start from the same prompt and initial latent. One path is conditioned only by the prompt, while the other path employs both the ID and the prompt as conditions. By semantically aligning the UNET features on these two paths, the model will learn how to embed ID without impacting the behavior of the original model. The overview of our method is shown in Fig. 2.

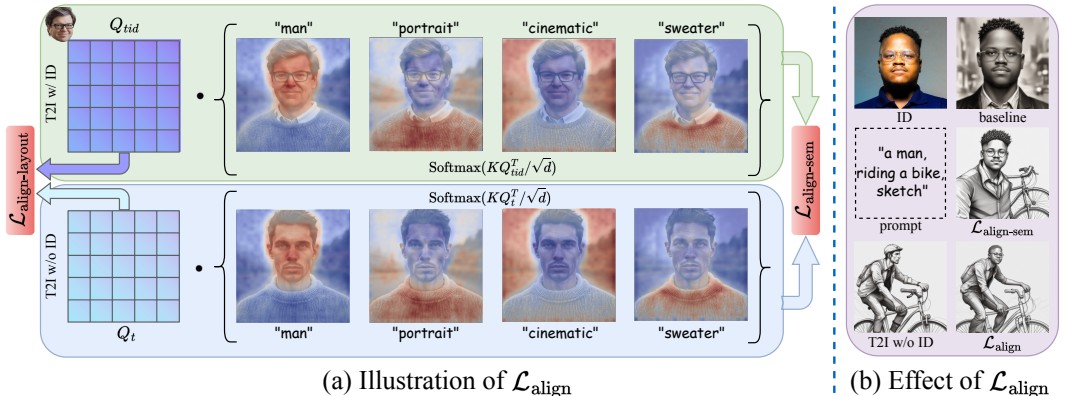

(a) Illustration of $\mathcal{L}_{\text{align}}$      (b) Effect of $\mathcal{L}_{\text{align}}$

Figure 3: Illustration and Effect of the alignment loss.

We chose to align the contrastive paths in their corresponding UNET's cross-attention layers. Specifically, we denote the UNET features in the path without ID embedding as $Q_t$, whereas the corresponding UNET features in the contrastive path with ID embedding as $Q_{tid}$. For simplicity, we omit the specific layers and denoising steps here. In actuality, alignment is conducted across all layers and time steps.

Our alignment loss consists of two components: the semantic alignment loss and the layout alignment loss. An illustration is presented in Fig. 3 (a). We use textual features $K$ to query the UNET features $Q$. For each token in $K$, it will calculate the correlation with $Q$, and further aggregate $Q$ based on the correlation matrix. Analogous to Eq. 2, the attention mechanism here can be expressed as $\text{Attention}(K, Q, Q)$, which can be interpreted as the response of the UNET features to the prompt. The insight behind our semantic alignment loss is simple: if the embedding of ID does not affect the original model's behavior, then the response of the UNET features to the prompt should be similar in both paths. Therefore, our semantic alignment loss $\mathcal{L}_{\text{align-sem}}$ can be formulated as follows:

$$\mathcal{L}_{\text{align-sem}} = \left\| \text{Softmax}\left(\frac{KQ_{tid}^T}{\sqrt{d}}\right)Q_{tid} - \text{Softmax}\left(\frac{KQ_t^T}{\sqrt{d}}\right)Q_t \right\|_2 . \tag{3}$$

As illustrated in Fig. 3 (b), the introduction of $\mathcal{L}_{\text{align-sem}}$ significantly mitigates the issue of ID information contaminating the model's behavior. However, it cannot guarantee layout consistency, so we add a layout alignment loss $\mathcal{L}_{\text{align-layout}}$, which is defined as:

$$\mathcal{L}_{\text{align-layout}} = \|Q_{tid} - Q_t\|_2 . \tag{4}$$

The full alignment loss is formulated as

$$\mathcal{L}_{\text{align}} = \lambda_{\text{align-sem}} \mathcal{L}_{\text{align-sem}} + \lambda_{\text{align-layout}} \mathcal{L}_{\text{align-layout}}, \tag{5}$$

where $\lambda_{\text{align-sem}}$ and $\lambda_{\text{align-layout}}$ serve as hyperparameters that determine the relative importance of each loss item. In practice, we set $\lambda_{\text{align-layout}}$ to a relatively small value, as we found that a larger value compromises the ID fidelity.

### 3.5 Optimizing ID Loss in a More Accurate Setting

In ID Customization tasks, ensuring a high degree of ID fidelity is essential, given our innate human sensitivity towards discerning facial features. To improve the ID fidelity, aside from enhancements on the ID encoder [50, 44, 53], another universal and parallel improvement is the introducing of an ID loss [6, 3, 29] during the training. However, these methods directly predict $x_0$ at the $t$-th timestep in the diffusion training process, only using a single step. This will produce a noisy and flawed predicted $x_0$, subsequently leading to inaccurate calculation of ID loss. To ease this issue, recent work [29] proposes to only applying the ID loss on less noisy stages. However, since the ID loss only affects a portion of timesteps, which may potentially limit the full effectiveness of it. In this study, thanks to the introduced Lightning T2I branch, the above issue can be fundamentally resolved. Firstly, we can swiftly generate an accurate $x_0$ conditioned on the ID from pure noise within 4 steps. Consequently, calculating the ID loss on this $x_0$, which is very close to the real-world data distribution, is evidently

more precise. Secondly, optimizing ID loss in a setting that aligns with the testing phase, is more direct and effective. Formally, the ID loss $\mathcal{L}_{id}$ is defined as:

$$\mathcal{L}_{\text{id}} = 1 - CosSim\left(\phi(C_{id}), \phi(\text{L-T2I}(x_T, C_{id}, C_{txt}))\right),$$ (6)

where $x_T$ denotes the pure noise, L-T2I represents the Lightning T2I branch, and $\phi$ denotes the face recognition backbone [6]. To generate photo-realistic faces, we fix the prompt $C_{txt}$ to "`portrait, color, cinematic`".

### 3.6 Full Objective

The full learning objective is defined as:

$$\mathcal{L} = \mathcal{L}_{\text{diff}} + \mathcal{L}_{\text{align}} + \lambda_{\text{id}}\mathcal{L}_{\text{id}}.$$ (7)

During training, only the newly introduced MLPs and the learnable linear layers $K_{id}$ and $V_{id}$ in cross-attention layers are optimized with this objective, with the rest remaining frozen.

## 4 Experiments

### 4.1 Implementation Details

We build our PuLID model based on SDXL [30] and the 4-step SDXL-Lightning [23]. For the ID encoder, we use antelopev2 [6] as the face recognition model and EVA-CLIP [40] as the CLIP Image encoder. Our training dataset comprises 1.5 million high-quality human images collected from the Internet, with captions automatically generated by BLIP-2 [20]. Our training process consists of three stages. In the first stage, we use the conventional diffusion loss $\mathcal{L}_{\text{diff}}$ to train the model. In the second stage, we resume from the first stage model and train with the ID loss $\mathcal{L}_{\text{id}}$ (we use arcface-50 [6] to calculate ID loss) and diffusion loss $\mathcal{L}_{\text{diff}}$. This model strives for the maximum ID fidelity without considering the contamination to the original model. In the third stage, we add the alignment loss $\mathcal{L}_{\text{align}}$ and use the full objective as shown in Eq. 7 to fine-tune the model. We set the $\lambda_{\text{align-sem}}$ to 0.6, $\lambda_{\text{align-layout}}$ to 0.1, and $\lambda_{\text{id}}$ to 1.0. In the Lightning T2I training branch, we set the resolution of the generated image to $768 \times 768$ to conserve memory. Training is performed with PyTorch and diffusers on 8 NVIDIA A100 GPUs in an internal cluster.

### 4.2 Test Settings

For consistency in comparison, unless otherwise specified, all the results in this paper are generated with the SDXL-Lightning [23] base model over 4 steps using the DPM++ 2M sampler [17]. The CFG-scale is set to 1.2, as recommended by [23]. Moreover, for each comparison sample, all methods utilize the same seed. We observe that the comparison methods, namely InstantID [44] and IPAdapter (more specifically, IPAdapter-FaceID [1]) are highly compatible with the SDXL-Lightning model. As shown in appendix subsection 7.5, when compared to using SDXL-base [30] as the base model, employing SDXL-Lightning results in InstantID generating more natural and aesthetically pleasing images, and enables IPAdapter to achieve higher ID fidelity. Furthermore, we provide a quantitative comparison with these methods on SDXL-base, with the conclusions remaining consistent with those on SDXL-Lightning.

To more effectively evaluate these methods, we collected a diverse portrait test set from the internet. This set covers a variety of skin tones, ages, and genders, totaling 120 images, which we refer to as DivID-120. As a supplementary resource, we also used a recent open-source test set, Unsplash-50 [8], which comprises 50 portrait images uploaded to the Unsplash website between February and March 2024.

### 4.3 Qualitative Comparison

As shown in Fig. 4, when compared to SOTA methods such as IPAdapter and InstantID, our PuLID tends to achieve higher ID fidelity while creating less disruption to the original model. From columns 1, 2, 5, 6, and 7, it is clear that our method can attain high ID similarity in realistic portrait scenes and delivers better aesthetics. Conversely, other methods either fall short in ID fidelity or show diminished aesthetics compared to the base model. Another distinct advantage of our approach is

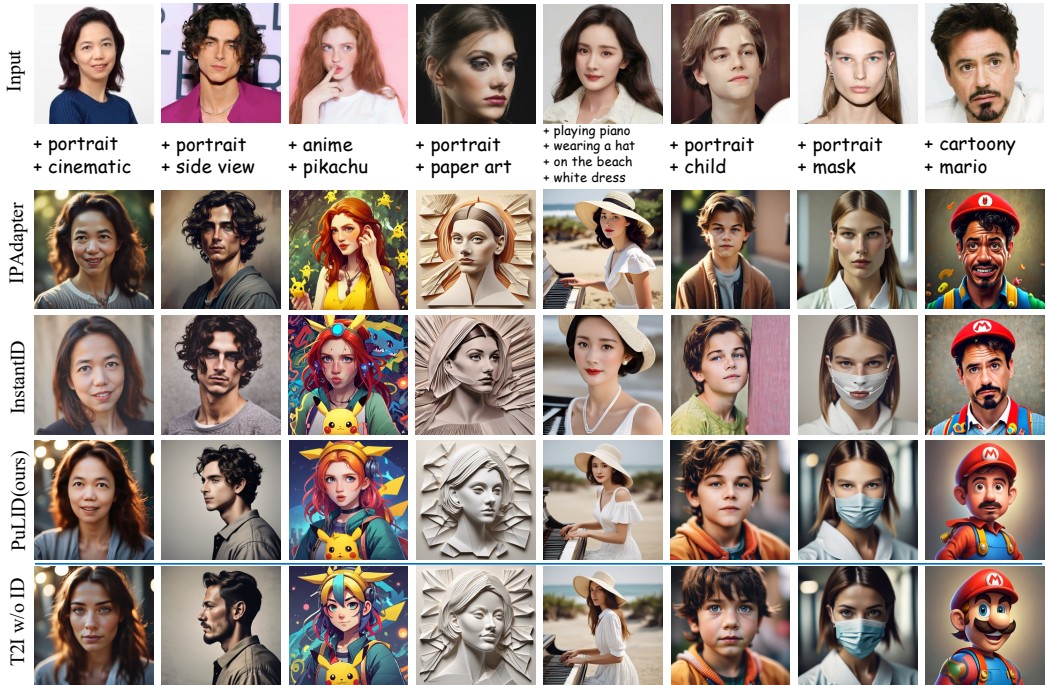

Figure 4: **Qualitative comparisons.** T2I w/o ID represents the output generated by the original T2I model without inserting ID, which reflects the behavior of the original model. Our PuLID achieves higher ID fidelity while causing less disruption to the original model. As the disruption to the model is reduced, results generated by PuLID accurately reproduce the lighting (1st row), style (4th row), and even layout (5th row) of the original model. This unique advantage broadens the scope for a more flexible application of PuLID.

Table 1: **Quantitative comparisons.** *We observed that PhotoMaker shows limited compatibility with SDXL-Lightning, hence, we compare its performance on SDXL-base in this table.

|  | DivID-120 | | | Unsplash-50 | | |
|---|---|---|---|---|---|---|
|  | Face Sim.↑ | CLIP-T↑ | CLIP-I | Face Sim.↑ | CLIP-T↑ | CLIP-I↑ |
| PhotoMaker* | 0.271 | 26.06 | 0.649 | 0.193 | 27.38 | 0.692 |
| IPAdapter | 0.619 | 28.36 | 0.703 | 0.615 | 28.71 | 0.701 |
| InstantID | 0.725 | 28.72 | 0.680 | 0.614 | 30.55 | 0.736 |
| PuLID (ours) | **0.733** | **31.31** | **0.812** | **0.659** | **32.16** | **0.840** |

that as the disruption to the model decreases, the results produced by PuLID accurately replicate the lighting (1st column), style (4th column), and even layout (5th column) of the original model. In contrast, although comparative methods can also perform stylization, notable style degradation can be noticed when compared to the original model. Finally, our model also possesses respectable prompt-editing capabilities, such as changing orientation (2nd column), altering attributes (6th column), and modifying accessories (7th column). **More qualitative results can be found in subsection 7.1 and subsection 7.2 of the appendix**.

### 4.4 Quantitative Comparison

The quantitative results are reported in Table 1. **Face Sim.** represents the ID cosine similarity, with ID embeddings extracted using CurricularFace [15]. CurricularFace is different from the face recognition models we use in the ID encoder and for calculating ID loss. **CLIP-T** [32] measures the ability to follow prompt. We also use **CLIP-I** to quantify the CLIP image similarity between two images before and after the ID insertion. A higher CLIP-I metric indicates a smaller modification in image elements (such as the background, composition, style) after ID insertion, suggesting a lower degree

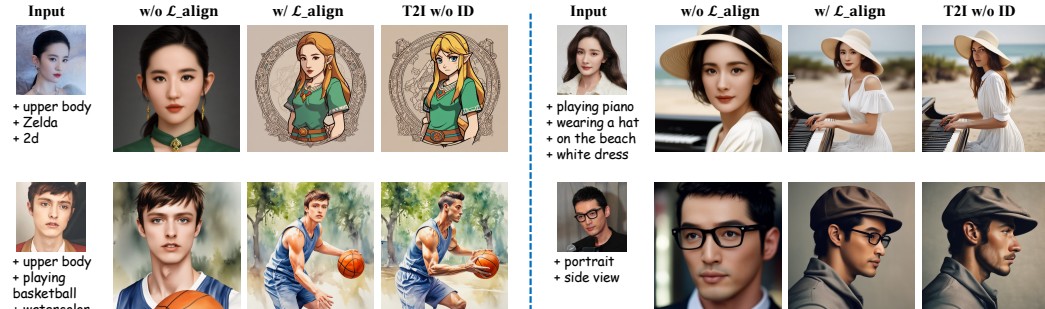

Figure 5: **Qualitative comparison for ablation study on alignment loss.**

Table 2: **Quantitative comparisons for ablation studies on ID loss and alignment loss.**

|  | DivID-120 | | | Unsplash-50 | | |
|---|---|---|---|---|---|---|
|  | Face Sim.↑ | CLIP-T↑ | CLIP-I↑ | Face Sim.↑ | CLIP-T↑ | CLIP-I↑ |
| Baseline (Stage1) | 0.561 | 29.06 | 0.736 | 0.514 | 30.16 | 0.769 |
| w/ ID Loss naive | 0.652 | 27.05 | 0.683 | 0.601 | 28.00 | 0.707 |
| w/ ID Loss (Stage2) | **0.761** | 24.91 | 0.624 | **0.708** | 25.83 | 0.646 |
| PuLID (Stage3) | 0.733 | **31.31** | **0.812** | 0.659 | **32.16** | **0.840** |

of disruption to the original model's behavior. We provide the evaluation prompts in the appendix. As observed from Table 1, our method, PuLID, surpasses comparison methods across all three metrics, achieves SOTA performance in terms of both ID fidelity and editability. Furthermore, our method significantly outperforms others with respect to the CLIP-I metric, implying that our method incurs much less intrusion on model behavior compared to other methods.

### 4.5 Ablation

**Alignment loss ablation**. Fig. 5 displays a qualitative comparison between models trained with and without the alignment loss $\mathcal{L}_{align}$. As observed, without $\mathcal{L}_{align}$, the embedding of ID severely disrupts the behavior of the original model. This disruption manifests as an inability for the prompt to precisely modify style (the left two cases of Fig. 5) and orientation (the lower right case of Fig. 5). Also, the layout would collapse to the extent that the face occupies the majority of the image area, resulting in a diminished diversification of the layout. However, with the introduction of our alignment loss, this disruption can be significantly reduced. From Table 2, we could also observe a large improvement in CLIP-T and CLIP-I when equipped with $\mathcal{L}_{align}$ (from Stage2 to Stage3).

**ID loss ablation**. Table 2 illustrates the improvement in ID fidelity using the naive ID loss (directly predicting $x_0$ from current timestep) and the more accurate ID loss $\mathcal{L}_{id}$ introduced in this paper, in comparison to the baseline. As observed, $\mathcal{L}_{id}$ can accomplish a greater improvement compared to the naive ID loss. We attribute this to the more precise $x_0$ provided by the Lightning-T2I branch, which also better aligns with the testing setting, thereby making the optimization of ID loss more direct and effective. Another worth mentioning point is that the baseline, naively trained on the internal dataset, underperforms in ID fidelity and editability. Conversely, the introduction of the PuLID training paradigms delivers significant enhancement. Therefore, this substantiates that the improvement mainly comes from the method, rather than the dataset.

## 5 Limitation

While our PuLID achieves superior ID fidelity and editability with minimal disruption to the base model's behavior, it still has limitations. Due to the incorporation of the Lightning T2I branch, the training speed per iteration is slower than that of conventional diffusion training, and it also demands more CUDA memory. However, this issue can be significantly mitigated when future fast sampling

methods can generate satisfying results in a single step (currently, we use 4 steps). Another limitation is that although the accurate ID loss introduced in the Lightning T2I branch markedly enhances ID fidelity, it also impacts image quality to some extent, such as causing blurriness in faces. This can be discerned in Fig. 3 (b) and Fig. 5. Nonetheless, this issue is not uniquely associated to our method, as similar phenomena has been observed in reward tuning methods [49, 31, 5, 47], with [5] attributing it to the reward-hacking problem. Although this issue can be largely alleviated by pairing with our proposed contrastive alignment loss, future work could explore a more effective ID loss that does not negatively affect image quality.

## 6 Conclusion

This paper presents PuLID, a novel tuning-free approach to ID customization in text-to-image generation. By incorporating a Lightning T2I branch along with a contrastive alignment strategy, PuLID achieves superior ID fidelity and editability with minimal disruption to the base model's behavior. Experimental results demonstrate that PuLID surpasses current methods, showcasing its potential for flexible and efficient personalized image generation. Future work could explore the application of this proposed training paradigm to other image customization tasks, like IP and style customization.

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

# 7 Appendix

## 7.1 More Applications of PuLID

We provide more applications of our PuLID in Fig. 6, encompassing style alterations (1st row), IP fusion (2nd row), accessories modification (3rd row), recontextualization (4th row), attributes editing (5th row), transformation from non-photo-realistic domains to photo-realistic ones (6th row), and ID mixing (7th row).

## 7.2 Generalization Ability of PuLID

**Testing With Community Models.** During training, we keep the base model parameters fixed. We use SDXL in the initial stage and SDXL-Lightning, which is distilled from SDXL, in the final two stages. SDXL-Lightning is effective in retaining the style and layout of SDXL, allowing our model to generalize to community models based on SDXL during testing. However, disruptions caused by ID insertion are more noticeable when testing other base models rather than SDXL-Lightning, our training base model. To combat this, we enhance compatibility by simultaneously aligning with multiple base models during training. This process involves loading different base models onto different GPU ranks, while a single ID adapter is shared across all ranks [24]. Using this strategy, our PuLID model attains robust compatibility with popular community models, both accelerated and non-accelerated ones. This includes models like RealVisXL (Fig.9), Juggernaut-XL-Lightning (Fig.10), and DreamShaper-XL-Lightning (Fig. 11).

**Training With Non-accelerated Base Models.** We want to highlight that the core essence of our paper is to introduce a more accurate ID loss and alignment loss in the T2I training branch to achieve better ID fidelity and editability. The fast sampling method (such as SDXL-Lightning) serves as an optional acceleration trick, but it is not indispensable. Without the fast sampling method, we would need 30 inference steps with CFG on the T2I branch, compared to the current 4 required inference steps without CFG. Due to CUDA memory bottleneck (we exclude the use of gradient checkpointing due to its significant speed penalty), it is not feasible to perform backpropagation (BP) of the gradient at all timesteps. Nonetheless, it remains possible to make optimization viable with strategic techniques. Particularly, for the optimization of ID loss, BP of the gradient happens only for the last few timesteps [5]. For the optimization of alignment loss, a timestep is randomly selected for BP of the gradient. Table 3 shows the differences in speed and memory consumption between training with and without acceleration. From the table, we can see that if we do not use fast sampling and take SDXL as the base model for the T2I training branch, efficiency will indeed be much lower. However, thanks to the carefully designed optimization strategies mentioned above, the training method presented in this paper can be effectively adapted to non-accelerated models, with performance being further improved, shown in Table 4. Additional visual results can be found in Fig. 12. In summary, our method does not rely on accelerated base models, thus reflecting the universality of our approach. We also successfully adapt PuLID to a much larger and non-accelerated base model, FLUX[2], and open-source it at https://github.com/ToTheBeginning/PuLID.

Table 3: **Speed and memory comparison between training with and without acceleration**. ts denotes timestep.

|  | BP last 1 ts | BP last 2 ts | BP last 3 ts | BP last 4 ts | BP last 20 ts |
|---|---|---|---|---|---|
| w/ fast sampling | 2.6s/iter(41GB) | 2.9s/iter(49GB) | 3.1s/iter(56GB) | 3.3s/iter(63GB) | - |
| w/o fast sampling | 6.6s/iter(50GB) | 7.0s/iter(65GB) | 7.3s/iter(80GB) | OOM | OOM |

Table 4: **Quantitative comparison between training with and without acceleration.**

|  | DivID-120 | | | Unsplash-50 | | |
|---|---|---|---|---|---|---|
|  | Face Sim.↑ | CLIP-T↑ | CLIP-I↑ | Face Sim.↑ | CLIP-T↑ | CLIP-I↑ |
| w/ fast sampling | 0.733 | 31.31 | 0.812 | 0.659 | 32.16 | 0.840 |
| w/o fast sampling | 0.743 | 31.75 | 0.842 | 0.687 | 32.58 | 0.865 |

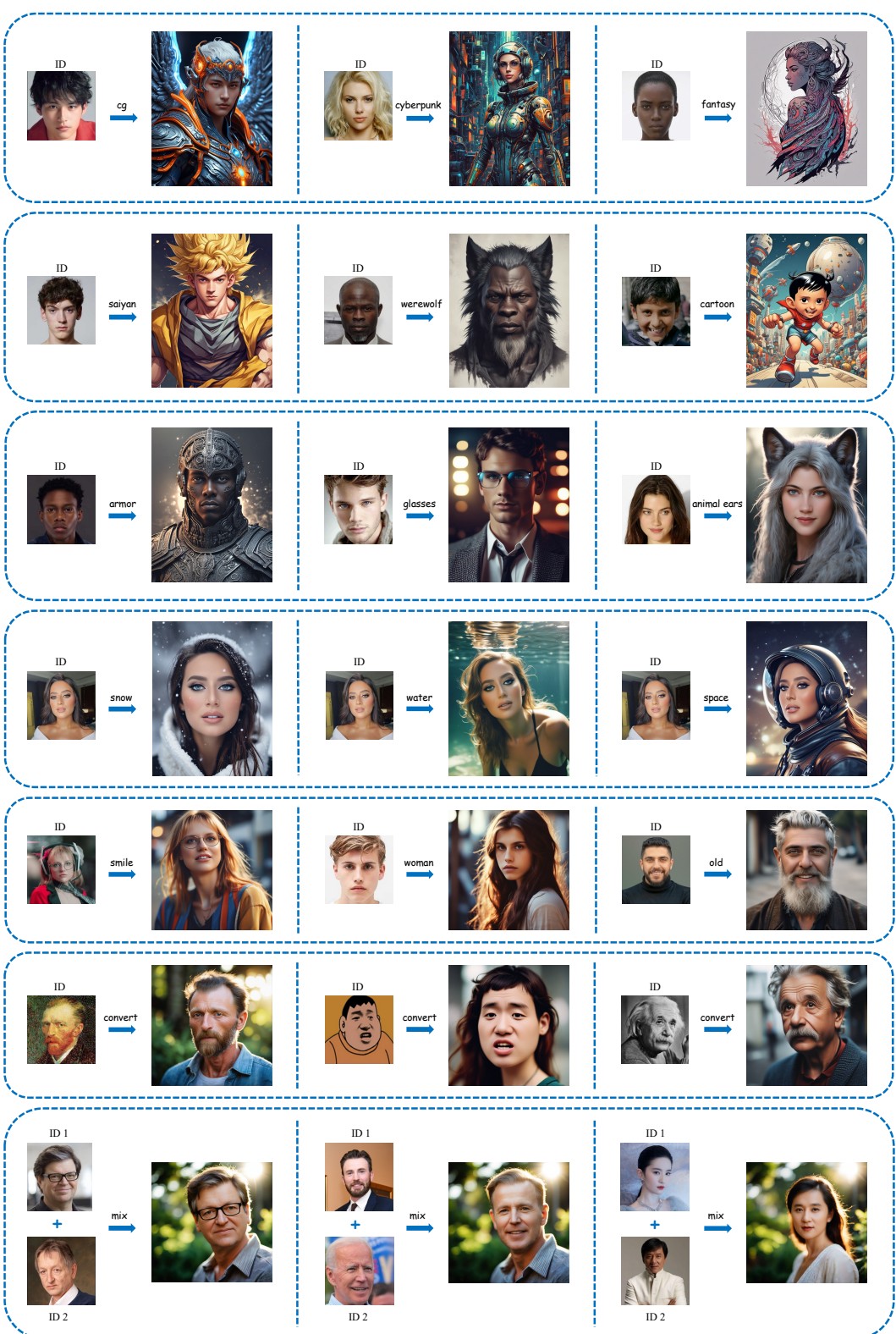

Figure 6: **More applications.** Including style changes, IP fusion, accessory modification, recontextualization, attribute editing, transformation from non-photo-realistic domain to photo-realistic domain, and ID mixing. Note that all these high-quality images are generated in just 4 steps with SDXL-Lightning model, without the need for additional Lora.

Table 5: **Quantitative comparison of training with different fast sampling methods and inference steps.**

| | DivID-120 | | | Unsplash-50 | | |
|---|---|---|---|---|---|---|
| | Face Sim.↑ | CLIP-T↑ | CLIP-I↑ | Face Sim.↑ | CLIP-T↑ | CLIP-I↑ |
| Hyper-SD T=1 | 0.694 | 31.91 | 0.819 | 0.632 | 31.89 | 0.857 |
| Hyper-SD T=2 | 0.720 | 32.08 | 0.810 | 0.653 | 32.35 | 0.840 |
| Lightning T=4 | 0.733 | 31.31 | 0.812 | 0.659 | 32.16 | 0.840 |
| Lightning T=8 | 0.734 | 31.66 | 0.818 | 0.668 | 32.19 | 0.850 |

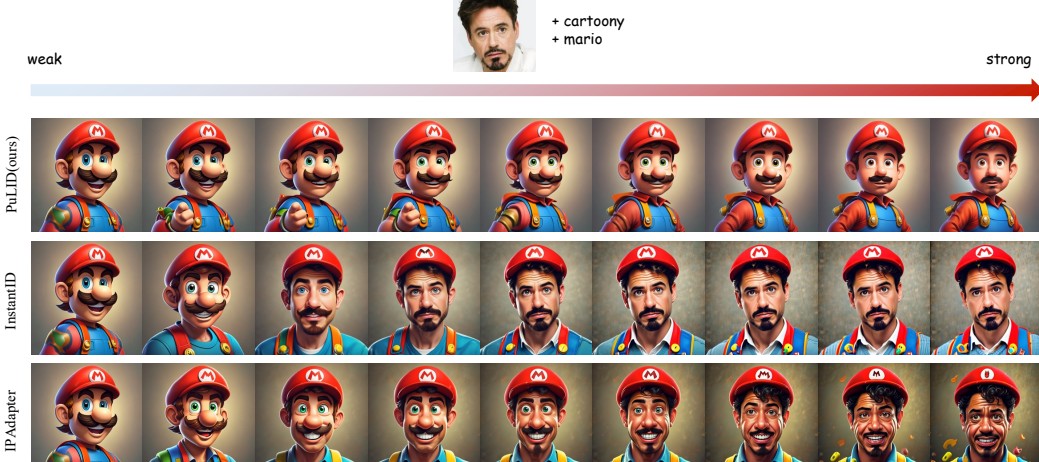

Figure 7: **Results under different ID weights.** The weights incrementally increase from left to right, culminating at the chosen weight at the far right.

### 7.3 Ablation Study on Fast Sampling Methods

Our selection criteria for a fast sampling method is that it can generate high-quality images within a limited number of steps while well preserving the style and layout of the original model. SDXL-Lightning fulfills these requirements and was the SOTA option at that time, so we selected it to accelerate our T2I training branch. Moreover, we settle on 4 steps as it achieves a balance between efficiency and quality. With fewer steps (*e.g.*, 1 or 2), the likelihood that SDXL-Lightning generates flawed faces increases. However, we have noted a recently emerged fast sampling method, Hyper-SD [34], which performs better than SDXL-Lightning on 1 and 2 steps. Therefore, we present the results of training on 1 step and 2 steps with Hyper-SD, as well as on 8 steps with SDXL-Lightning in Table 5. As shown in this table, training with 1 or 2 steps leads to a reduction in face similarity. Meanwhile, training with 8 steps slightly enhances overall performance. Considering both efficiency and performance, 4 steps is a sound choice.

### 7.4 More Details about the Test Settings

**Evaluation Prompts.** Table 7 presents the complete list of prompts utilized in the calculation of CLIP-T and CLIP-I. For measuring Face Sim., we employ the prompt "portrait, color, cinematic, in garden, soft light, detailed face" to guarantee that the generated face is photo-realistic and detectable. For each ID and each prompt, we randomly generate four images for evaluation.

**Hyperparameter Selection for the Comparison Methods.** We employ the popular and widely used ComfyUI workflow [2] in the community to test IPAdapter and InstantID. For InstantID, the weight is set to 0.8. For IPAdapter, the lora scale is set to 0.8 and the IP_weight is set to 1.5. Additionally, we present the results of these methods under different ID weights (as shown in Fig. 7). It can be

---

[2]https://github.com/cubiq/ComfyUI_InstantID ; https://github.com/cubiq/ComfyUI_IPAdapter_plus

Table 6: **Quantitative comparisons on SDXL-base.**

| | DivID-120 | | | Unsplash-50 | | |
|---|---|---|---|---|---|---|
| | Face Sim.↑ | CLIP-T↑ | CLIP-I↑ | Face Sim.↑ | CLIP-T↑ | CLIP-I↑ |
| PhotoMaker | 0.271 | 26.06 | 0.649 | 0.193 | 27.38 | 0.692 |
| IPAdapter | 0.597 | 29.39 | 0.720 | 0.572 | 30.03 | 0.749 |
| InstantID | 0.755 | 25.33 | 0.608 | 0.648 | 26.41 | 0.649 |
| PuLID (max. ID sim.) | **0.773** | 24.77 | 0.598 | **0.711** | 25.33 | 0.628 |
| PuLID (ours) | 0.722 | **30.69** | **0.781** | 0.654 | **31.23** | **0.813** |

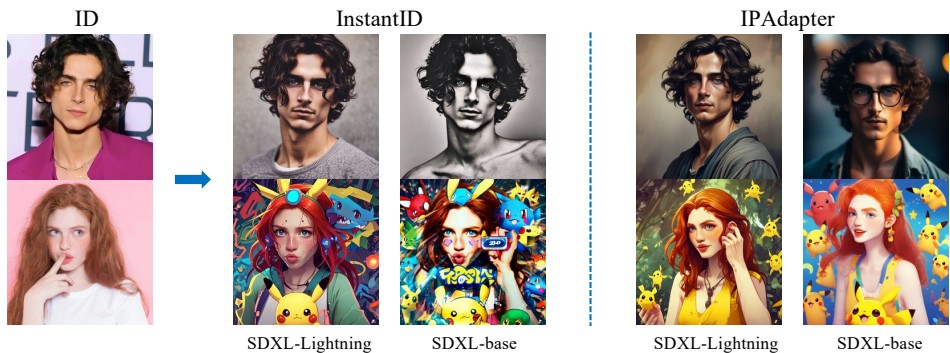

Figure 8: **Visual comparison of different base model for InstantID and IPAdapter.**

observed that for InstantID and IPAdapter, the ID embedding only has minimal impact on the original model when the ID weight is set to a relatively small value. However, at this point, the ID similarity decreases significantly. Since our method only influences the ID-relevant parts, it doesn't significantly disrupt the original model's behavior, even when increasing ID weight.

## 7.5 Comparisons on SDXL-base

**The Influence of Changing Base Model on the Comparison Methods.** Here, we present the visual differences associated with these methods when varying base models are used. Fig. 8 showcases the qualitative comparison results. As observable from the figure, InstantID attains better image quality and aesthetics when using SDXL-Lightning instead of SDXL-base, while the ID fidelity shows little visual difference. For IPAdapter, a better ID fidelity can be achieved using SDXL-Lightning compared to SDXL-base.

**Quantitative Comparison.** We provide the quantitative comparison on SDXL-base in Table. 6. As evident, our final model, PuLID, still outperforms the comparative methods in most scenarios, with the exception of Face Sim. on DivID-120, where our method slightly falls behind InstantID. Nevertheless, our method significantly surpasses InstantID in terms of CLIP-T and CLIP-I metrics, and our model (from Stage2) that strives for maximum ID similarity also manages to surpass InstantID on the Face Sim. metric. Another intriguing and noteworthy point is that typically, an enhancement in ID similarity accompanies a decrease in the CLIP-T and CLIP-I metrics. However, our method has the unique capability to not only achieve a high degree of ID similarity, but concurrently maintain high editability and low interference. This is facilitated by the novel training paradigm introduced in this study.

## 7.6 Prompt List for Contrastive Alignment

Table 8 presents the complete list of prompts used in contrastive alignment. Despite their simplicity, these prompts already ensure an effective methodology. Future exploration will involve determining if there exists a more optimal list of prompts.

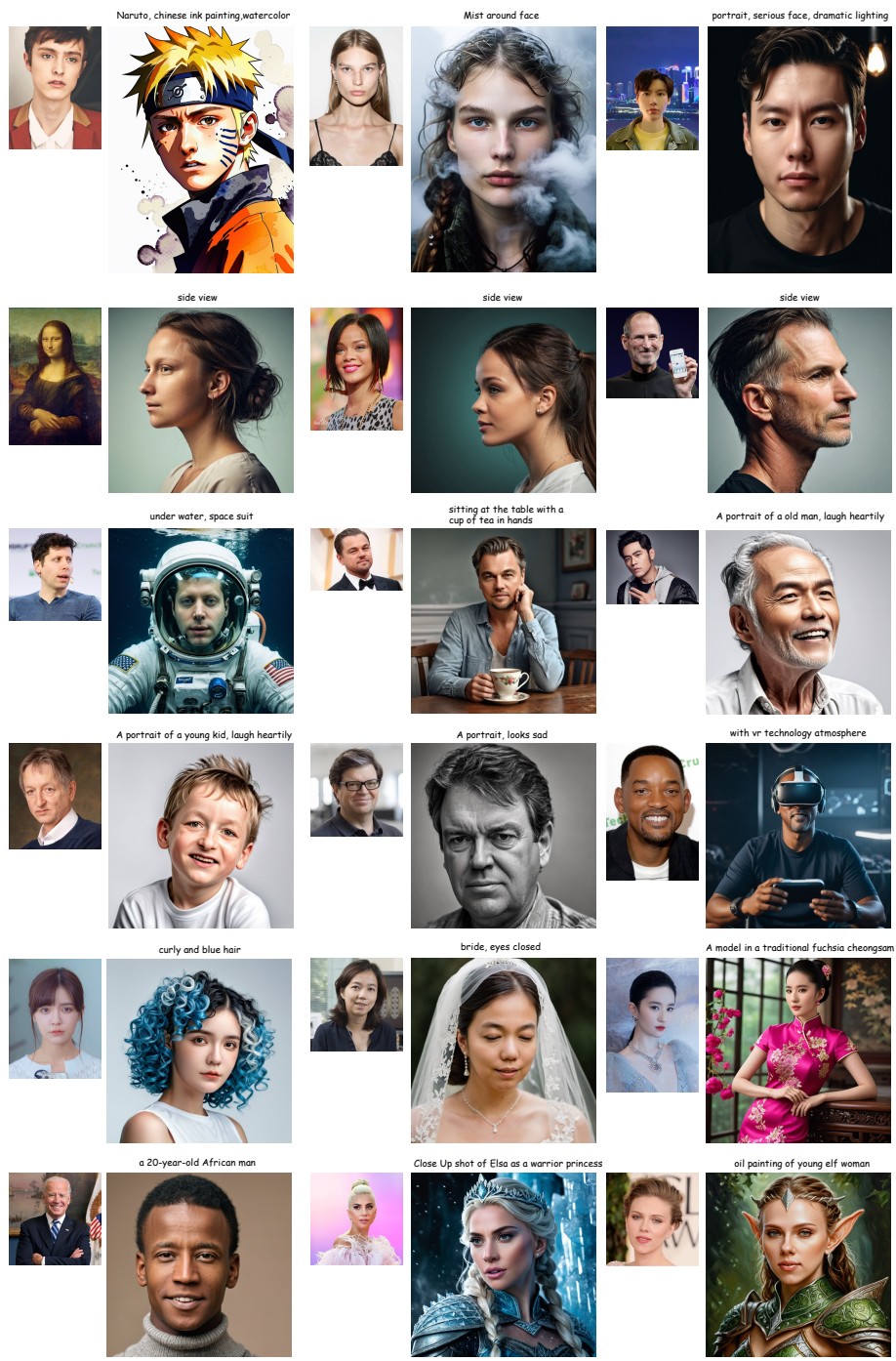

Figure 9: **PuLID with RealVis-XL as base model.** Zoom in for best view

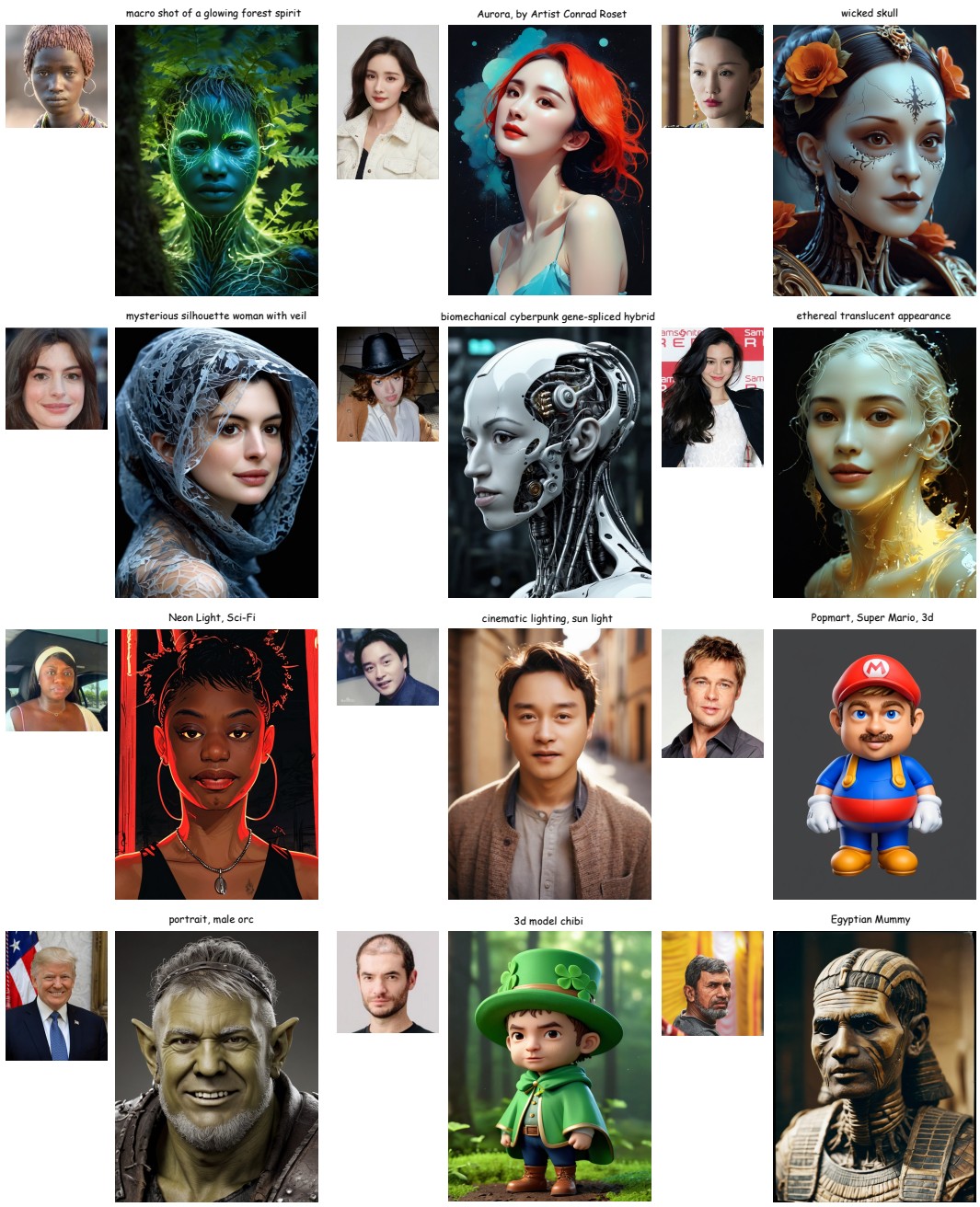

Figure 10: **PuLID with Juggernaut-XL-Lightning as base model.** Zoom in for best view

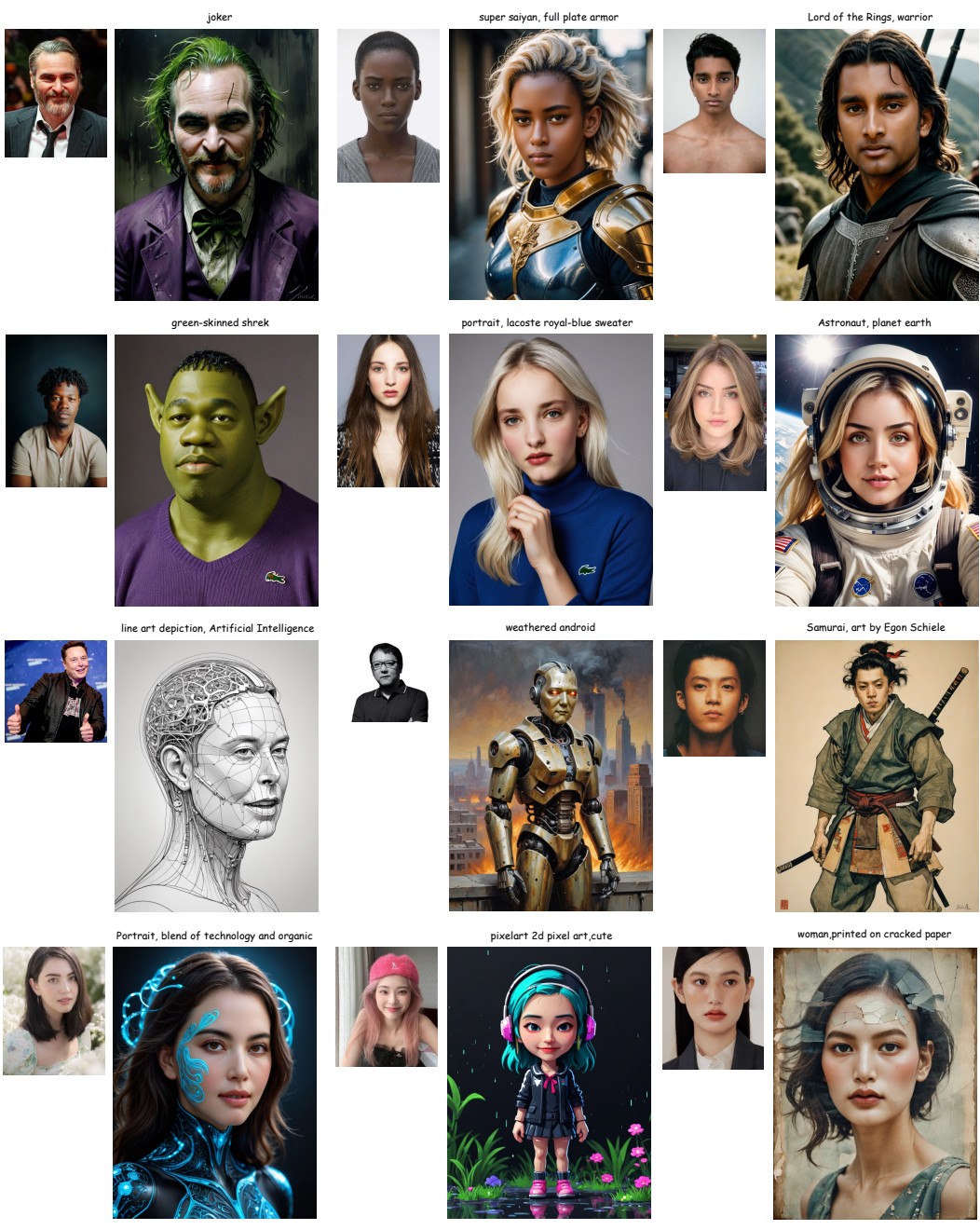

Figure 11: **PuLID with Dreamshaper-XL-Lightning as base model.** Zoom in for best view

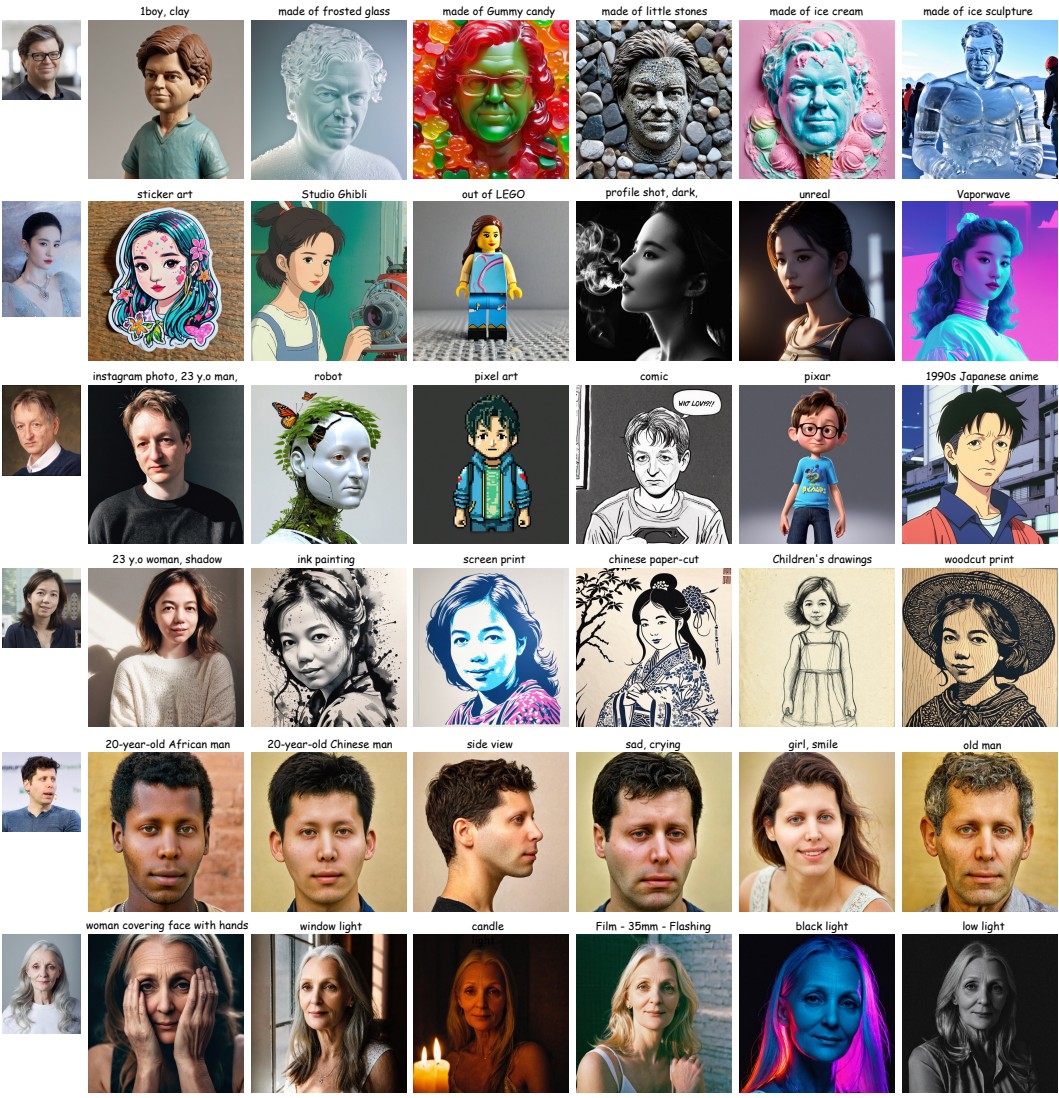

Figure 12: **Qualitative results on PuLID trained without fast sampling methods.** We test the model on a wide range of styles, face materials, lighting conditions, and editing tasks, demonstrating the effectiveness of generalizability of PuLID. Zoom in for best view.

## 7.7 Broader Impacts

Thanks to the superior ID fidelity and editability offered by PuLID, this technology holds the potential to enhance personalized content creation, offering benefits for various applications such as entertainment and virtual reality. However, it could also be exploited to create misleading or false representations of individuals, potentially posing privacy infringements or being used maliciously, such as in deepfakes. Although existing deepfake detection methods [27] can help mitigate these risks, we underscore the imperative role of developing and adhering to stringent ethical guidelines for ensuring responsible use of such technology.

| Category | Prompt |
|---|---|
| Clothing | portrait, a person wearing a spacesuit |
| Accessory | portrait, a person wearing a surgical mask
portrait, a person wearing an eye mask
portrait, a person wearing headphones with red hair
portrait, a person wearing a doctoral cap |
| Action | portrait, a person cooking in the kitchen
portrait, a person coding in front of a computer |
| Attribute | portrait, a young child laughing at the camera
portrait, an angry person,old |
| View | portrait, color,side view |
| background | portrait, with a beautiful purple sunset at the beach in the background |
| Style | portrait, pencil drawing
portrait,3D Animation, Disney style,cute,popmart blindbox
portrait, kawaii style, cute, adorable, brightly colored, cheerful, anime influence, highly detailed
portrait,latte art in a cup |
| Complex | portrait, side view, in papercraft style
portrait, a garden gnome, in papercraft sketch, wearing a glasses
portrait, Madhubani, wearing a mask
portrait, Ukiyo-e Painting style, playing the violin
portrait of green-skinned shrek, wearing lacoste purple sweater |

Table 7: **Evaluation prompts.**

- portrait, color, cinematic

- portrait, anime artwork

- portrait, comic art, graphic novel art

- portrait, digital artwork, illustrative, painterly, matte painting

- portrait, magnificent, celestial, ethereal, painterly, epic, majestic, magical, fantasy art, cover art, dreamy

- portrait, vibrant, professional, sleek, modern, minimalist, graphic, line art, vector graphics

- portrait, cyberpunk, vaporwave, neon, vibes, vibrant, stunningly beautiful, crisp, detailed, sleek, ultramodern, magenta highlights, dark purple shadows, high contrast, cinematic, ultra detailed, intricate, professional

- portrait, energetic brushwork, bold colors, abstract forms, expressive, emotional

- portrait, film noir style, monochrome, high contrast, dramatic shadows, 1940s style, mysterious, cinematic

- portrait, papercut, mixed media, textured paper, overlapping, asymmetrical, abstract, vibrant

- portrait, paper mache representation, 3D, sculptural, textured, handmade, vibrant, fun

- portrait, vaporwave style, retro aesthetic, cyberpunk, vibrant, neon colors, vintage 80s and 90s style, highly detailed

- portrait, watercolor painting, vibrant, beautiful, painterly, detailed, textural, artistic

- portrait, impressionist painting, loose brushwork, vibrant color, light and shadow play

- portrait, expressionist, raw, emotional, dynamic, distortion for emotional effect, vibrant, use of unusual colors, detailed

Table 8: **Prompt list for contrastive alignment.**

