# OpenReview forum: "PuLID: Pure and Lightning ID Customization via Contrastive Alignment"
_NeurIPS.cc/2024/Conference — NeurIPS 2024 poster_

### Official Review · Reviewer_pgk9 · 2024-07-09

**Soundness:** 3
**Presentation:** 4
**Contribution:** 4
**Rating:** 7
**Confidence:** 5

**Summary:**

In this paper, an ID customization method for text-to-image generation without tuning is studied. In practical applications, the method often encounters the problems of lack of ID fidelity and interference of the original pattern line by inserting ID. To solve this problem, the author proposes a new ID customization method for text-to-image generation without tuning. A large number of experiments, called Pure and Lightning ID customization (PuLID), on multiple benchmark datasets validate the superiority of the proposed PuLID.

**Strengths:**

The paper is well motivated, cleverly designed and clearly illustrated. The experimental results are obviously superior to other methods.

**Weaknesses:**

The author's work is very good, but I still have three questions I would like to discuss with them.

1. Experimental results show that PuLID has achieved good performance on specific data sets, but the generalization ability of the model on different types and styles of data sets has not been discussed much in this paper. Can the author provide more experimental results on the generalization of the model? In addition, is there further analysis and discussion of possible limitations of the model, such as its performance in handling certain ID features or style transitions?

2. The authors mentioned that in order to solve the two problems mentioned in their method, the final loss calculation is based on the combination of three kinds of losses. What impact do the different proportions of the three kinds of losses have on the experimental results, especially the impact of ID loss?

3.The paper points out that high-quality images can be generated in fewer steps through the Lightning T2I branch, and ID losses can be calculated on this basis. Is there a significant improvement in computing efficiency with this method? Compared to the traditional multi-step diffusion model, what is the training and reasoning time of the Lightning T2I branch in practical applications?

4.The PULID used by the author in the experiment is based on SDXL and the 4-step SDXL-Lightning. Is it possible to conduct experiments based on other diffusion models to judge its universality？

**Questions:**

I am mainly concerned about the universality of the proposed method and theoretical analysis, whether the proposed Lightning T2I branch can be applied to other diffusion models, and how fast and efficient it is in the actual training and reasoning process.

**Limitations:**

In general, the content and theory of the article are very rich, but there are some limitations. 1) There are few experimental data, which is not well reflected in aspects such as universality and speed. 2) Is it possible to extend this method and replace ID with other things, such as (clothing)?

---

> ### Author Rebuttal · Authors · 2024-08-06
>
> Thanks for your acknowledgment and careful reviews. Our responses are as follows:
>
> **`Q1:` Discuss the limitations and generalization ability of the method.**
>
> The limitations have been discussed in Section 6.1 of the paper. In terms of generalizability of PuLID, the paper has validated its effectiveness on the following grounds:
>
> 1. Section 6.5 presents the compatibility of the PuLID model on some widely-used community base models (both accelerated and non-accelerated models), exhibiting the broad applicability of our method.
>
> 2. We evaluate our method on two datasets. DivID-120, proposed in this paper, encapsulates diverse skin tones, ages, and genders, while Upsplash-50, proposed by LCM-Lookahead, comprises non-celebrity photos newly uploaded this year, hence unlikely to be a part of the training dataset. These datasets offer a wide range of input types. We will release DivID-120 to facilitate the evaluations for ID customization methods.
>
> 3. The qualitative results showcased in Figures 1, 4, 6, 9, 10, 11 demonstrate various style transition results, including 2D, 3D, anime, pixel, line art, cyberpunk, cg, fantasy, cartoon, etc., alongside a spectrum of facial edits. Most of these scenarios were not included in the training prompts, further underscoring our model's generalization abilities.
>
> We also provide more qualitative results in **Figure 1 of the Rebuttal PDF**.
>
> ---
>
> **`Q2:` How do the proportions of the three losses affect experimental results, especially ID loss?**
>
> - *Impact of diffusion loss*: In practice, we keep the weight of the diffusion loss constant (i.e., 1) while adjusting the weights for ID loss and alignment loss. In extreme experiments where we exclude the diffusion loss, the detailed facial fidelity slightly degrades, particularly with regard to eyewear and hairstyles. This is because the ID loss doesn't directly constrain these details, hence the reconstruction loss from the diffusion loss helps in preserving them.
>
> - *Impact of ID Loss and Alignment Loss*: The weights of these two losses determine the trade-off between ID similarity and editability. Increasing the weight of ID loss enhances ID similarity but reduces editability. Conversely, increasing the weight of alignment loss boosts editability but lowers ID similarity.
>
> ---
>
> **`Q3:` Is there a significant improvement in computing efficiency with fast sampling method, compared to the traditional multi-step (e.g., 30) diffusion model.**
>
> Please refer to `GQ1`. When compared to the traditional diffusion model's 30-step inference, using a 4-step fast sampling can make the training process twice as fast; if compared with a 50-step inference, it would be three times faster.
>
> ---
>
> **`Q4:` Concerns about the universality of this method.**
>
> As explained in `GQ1`, our method can be adapted to base models without acceleration plugins, demonstrating the method's generalizability and universality.
>
> ---
>
> **`Q5:` Is it possible to extend this method and replace ID with other things, such as (clothing)?**
>
> It is theoretically feasible; please refer to `GQ3` for more details.

---

> > ### Comment · Reviewer_pgk9 · 2024-08-09
> >
> > This is a good job, and the author has also answered my questions well.

---

> > > ### Author Response · Authors · 2024-08-09
> > >
> > > We thank Reviewer pgk9 for taking the time to review our response. We greatly appreciate your acknowledgment and are pleased that we could address your concerns.

---

### Official Review · Reviewer_BtJK · 2024-07-11

**Soundness:** 4
**Presentation:** 4
**Contribution:** 4
**Rating:** 8
**Confidence:** 4

**Summary:**

This paper introduces Pure and Lightning ID customization (PuLID). It is a tuning-free approach for customizing identities in text-to-image generation. The model is trained over a huge dataset.

Technically, it integrates a Lightning T2I branch alongside a standard diffusion model, incorporating contrastive alignment loss and accurate ID loss to maintain fidelity to the original model.

Experimental results show PuLID achieves superior performance in preserving identity fidelity and enhancing editability. Additionally, PuLID ensures consistency in image elements like background, lighting, composition, and style throughout the identity customization process.

**Strengths:**

1. This paper is well-written as it explores the shortcomings of previous approaches in realistic scenarios, highlighting issues such as fidelity and the impact of inserting IDs into original T2I diffusion models.

2. The novelty of this paper lies in its new perspective on training paradigms with the introduction of PuLID. Unlike existing ID-preservation papers that focus on ID encoders, PuLID distinguishes itself by incorporating a Lightning T2I branch. This distinction is further enhanced through the integration of ID loss and alignment loss.

3. The experiments conducted in this study demonstrate superior performance both quantitatively and qualitatively compared to current methods.

**Weaknesses:**

1. The primary limitation, in my opinion, is that the PuLID model is trained using an internal dataset. This raises questions about whether the observed improvements stem primarily from the introduction of this new dataset or from the efficacy of the method itself. Consequently, it's unclear whether comparisons with other methods on public datasets are fair or solely reflective of dataset differences.

2. Another picky question for the authors is to include more quantitative comparison metrics as in PhotoMaker[1] and ConsistentID[2], including DINO, Face Diversity, FID, Speed, FGIS (fine-grained identity similarity), etc. Furthermore, can you explain why the experimental results in your paper are much lower than that showed in the original PhotoMaker paper?

3. It seems the T2I lightning branch is pretty important for your method PuLID. Have you done any ablation study over the choices of the T2I lightning models, e.g. the LCM, SDXL-turbo, TCM, InstaFlow, UFOGen, etc? This is to verify your model can be based on SDXL or other backbones, and the lightning models are with diverse choices. Also you choose T=4, would it be influenced by T=1,2,8...?




[1] PhotoMaker: Customizing Realistic Human Photos via Stacked ID Embedding

[2] ConsistentID: Portrait Generation with Multimodal Fine-Grained Identity Preserving

**Questions:**

Check the weaknesses

**Limitations:**

Check the weaknesses

---

> ### Author Rebuttal · Authors · 2024-08-06
>
> Thanks for your constructive and positive comments. Our responses are as follows:
>
> **`Q1:` Concerns about the internal training dataset.**
>
> **1. Dataset or method – primary source of improvement?**
> Table 2 in the main paper illustrates that a baseline, naively trained on the internal dataset, underperforms in ID fidelity and editability. Conversely, the introduction of the PuLID training paradigms delivers significant enhancement. Therefore, this substantiates that the improvement mainly comes from the method, rather than the dataset. We will revise the table analysis to emphasize this point.
>
>
> **2. Fairness in comparing methods trained on different datasets?**
> Existing SOTA methods often utilize internal and private training datasets. For instance, InstantID uses a 60 million-entry dataset, compared to our 1.5 million, and PhotoMaker uses an internal ID-group dataset, more challenging to gather than a non-ID-group dataset.
>
> **3. Lower dataset quality requirements with our method.** We would like to highlight that our method is less dependent on high-quality training datasets. Particularly, the ID loss and alignment loss in the Lightning T2I branch only require uncaptioned facial regions. Further, the unique feature of PuLID, which minimize the contamination to the original model behavior, ensures that the results are unbiased towards the quality of the training dataset, effectively reducing dataset quality requirements.
>
> ---
>
> **`Q2:` More quantitative metrics.**
>
> Thanks for your suggestion, here are the results on additional metrics:
>
> |            | Speed↑   | FGIS↑ | FID↓ | Face Div.↑ |
> | ---------- | ---------- | ----- | ------ | ---------- |
> | PhotoMaker | 8.69iter/s | 0.596 | 147.62 | 0.531      |
> | IPAdapter  | 6.08iter/s | 0.571 | 139.33 | 0.536      |
> | InstantID  | 4.66iter/s | 0.592 | 152.28 | 0.502      |
> | PuLID      | 5.85iter/s | 0.619 | 89.80  | 0.527      |
>
> *Speed* is tested on an A100 GPU in the ComfyUI environment at a resolution of 1024x1024. Our method, PuLID, and IPAdapter have similar speeds, while InstantID is slower due to its inclusion of a ControlNet. PhotoMaker is the fastest because it uses LoRA in attention layers, rather than new to_k and to_v layers like adapter-based methods, thus facilitating faster inference speeds by fusing LoRA before testing.
>
> *FGIS* measures the similarity of DINO embeddings in the facial region. While our method outperforms other methods on this metric, we would also like to emphasize that the embeddings extracted by the backbone trained on face recognition are more aligned with human perception and are widely recognized.
>
> *FID* measures the distance between the distributions of generated images with and without ID insertion. Our method significantly outperforms others on this metric, indicating that our method causes less disruption to the base model after ID insertion.
>
> *Face Div.* measures the diversity of generated images by calculating the LPIPS scores between pairs of faces in the generated images. Our method is comparable to PhotoMaker and IPAdapter in this regard. However, it's worthy to note that we achieved such diversity under the condition of much higher face similarity.
>
> Please note that we do not provide the DINO similarity for the full image region as we believe that measuring DINO similarity across the entire image is less accurate and reasonable compared to focusing on the facial region.
>
>
> ---
>
> **`Q3:` Reasons for PhotoMaker's Underperformance in the Experimental Results.**
>
> The observed underperformance of PhotoMaker is anticipated and aligns with findings from concurrent research, namely LCM-Lookahead. We speculate that this may be attributed to the narrow scope of celebrities in PhotoMaker's training dataset, limiting its effectiveness when applied to non-celebrities, as discussed in L41-L43 of the main paper.
>
> ---
>
> **`Q4:` Ablation on different fast sampling methods and different steps.**
>
> Please refer to `GQ2` for detailed comparison.

---

> > ### Comment · Reviewer_BtJK · 2024-08-12
> > **Thanks for the rebuttal**
> >
> > My concerns are partially addressed in the rebuttal. I prefer to see this project open source and more people engage to make it function better and more effective. Here I prefer to keep my rating.

---

> > > ### Author Response · Authors · 2024-08-12
> > >
> > > We thank Reviewer BtJK for taking the time to review our response. We highly agree with your suggestion of open-sourcing, which aligns with our original intentions. Furthermore, we plan to adapt PuLID to FLUX.1 in the coming months and make it open-source as well.

---

### Official Review · Reviewer_GW6B · 2024-07-18

**Soundness:** 3
**Presentation:** 3
**Contribution:** 2
**Rating:** 7
**Confidence:** 3

**Summary:**

This article presents a novel fine-tuning-free ID customization method called PuLID for text-to-image generation tasks. The method introduces Lightning T2I and contrast alignment loss, aiming to minimize the interference with the original model behavior while maintaining high ID fidelity. Experiments show that PuLID achieves excellent performance in terms of both ID fidelity and editability.

**Strengths:**

1. Introducing Lightning T2I branching and contrast learning approaches to achieve better performing ID customization methods.
2. The authors provide a wealth of ablation experiments to demonstrate the effectiveness of the method.

**Weaknesses:**

Here are some points and suggestions on how the authors should address them:

Clarification of Contributions:
The authors should clearly delineate their contributions in the manuscript. If the primary contribution is the use of contrastive alignment loss and ID loss, they should emphasize how these methods specifically address the problem of high ID similarity and reduce ID interference.
They should provide a more detailed explanation of how these losses are implemented and why they are effective.

Related Work:
The authors should expand their discussion on related work to include whether contrastive learning has been applied in similar contexts. This would help situate their work within the broader field and demonstrate its novelty.

Method Section:
There should be a more detailed explanation of how the method was realized. This includes a clear description of the steps taken, the rationale behind the chosen approach, and any unique aspects of the implementation.

Theoretical Analysis:
The authors should provide a more in-depth theoretical analysis, particularly for components like L_align-sem. They should explain the theoretical underpinnings of their approach and how it contributes to the overall effectiveness of the method.
Discussion on Main Contributions:

The manuscript should include a richer discussion on the main contributions. For example, the authors should explore whether the proposed loss functions reduce the weight of ID in text-to-image generation tasks. They should provide evidence or reasoning to support their claims.

Ablation Studies:
The authors should conduct ablation experiments to validate the necessity and effectiveness of each component of their method. For instance, if L_align-layout is introduced without ablation studies, it's unclear how crucial this component is to the overall performance.
Experimental Results:

While the effectiveness is illustrated in the experimental results, the authors should ensure that these results are robust and that the experiments are designed to test the method under various conditions.

Layout Consistency:
The authors should address the issue of layout consistency more thoroughly. If L_align-layout is mentioned but not rigorously tested, they should either conduct additional experiments or provide a theoretical justification for its inclusion.
Overall Clarity and Depth:

The authors should ensure that the manuscript is clear and that the depth of the discussion matches the complexity of the problem. This includes explaining the limitations of their approach and how it compares to existing methods.
By addressing these points, the authors can provide a more comprehensive and convincing argument for their contributions, making their research more accessible and impactful to the academic community.

**Questions:**

1. Lines 133 and 195 both mention that Q is an image feature of UNet, could the authors please explain how that image feature was obtained.
2. Is L_align-layout necessary? Is there any connection to your solution to the problem of ID information polluting the prompt?
3. In addition I would like to know how the authors' approach relates to the CFG scale (Classifier-Free Guidance scale) and whether CFG is valid for this task.

**Limitations:**

YES

---

> ### Author Rebuttal · Authors · 2024-08-06
>
> Thanks for your helpful advices. We respond to your core questions as follows:
>
>
> **`Q1:` Lines 133 and 195 both mention that Q is an image feature of UNet, please explain how that image feature was obtained.**
>
> In each cross-attention layer of UNET, the UNET image features are projected into query features via a linear layer. We denote these features as Q. This is reflected in L132-L134 of the main paper. We will revise the description to make this clearer.
>
> ---
>
> **`Q2:` Is L_align-layout necessary? Is there any connection to your solution to the problem of ID information polluting the prompt?**
>
> Yes, it is necessary. We consider layout and composition of the generated images as part of the original model's behavior. Therefore, if the layout and composition change after ID insertion, we perceive it as a disruption to the original model's behavior. The L_{align-layout} is helpful in maintaining the original model's layout and composition, as demonstrated in Figure 3 (b) of the paper, we also provide more qualitative results in **Figure 2 of the Rebuttal PDF**. For a quantitative ablation, we provide the table below:
>
> |                             | Face Sim. | CLIP-T | CLIP-I |
> | --------------------------- | --------- | ------ | ------ |
> | Stage2                      | 0.761     | 24.91  | 0.624  |
> | Stage3 w/o L_{align-layout} | 0.728     | 30.33  | 0.758  |
> | Stage3                      | 0.733     | 31.31  | 0.812  |
>
> As observed from the table, if L_{align-layout} is removed in stage3, despite achieving the same level of face similarity as stage3 with all losses, both CLIP-T and CLIP-I decrease. While the decline in CLIP-T is relatively mild, the decrease in CLIP-I is more significant, as it tends to reflect consistency in layout and composition to a certain extent.
>
>
> ---
>
> **`Q3:` How the approach relates to the CFG scale and whether CFG is valid for this task**
>
> During training, our Lightning T2I branch does not require CFG, hence the CFG scale is 1. However, our method can incorporate CFG during training. When the SDXL-Lightning acceleration is not employed (refer to GQ1), a CFG scale of 7.5 is used.
>
> As for testing, various CFG scales are compatible with our approach. For the base models utilized in our paper, we adopt the recommended CFG scales, such as 1.2 for SDXL-Lightning, 7 for RealVisXL, and 2 for Juggernaut-XL-Lightning.

---

### Official Review · Reviewer_bfEK · 2024-07-30

**Soundness:** 3
**Presentation:** 2
**Contribution:** 2
**Rating:** 7
**Confidence:** 2

**Summary:**

The paper propose a tuning-free method for customization text-to-image diffusion model. Particularly, the author propose to utilize efficient diffusion model (SDXL-lightning in this case) to generate samples during training. Then, they adopt contrastive alignment loss to preserve the identity of subject in the input.

**Strengths:**

- The paper is fairly well-written and easy to read.
- The proposed method is well-motivated and sound.
- The ablation experiment demonstrates the effectiveness of each proposed components.
- Reported results seem quite impressive.

**Weaknesses:**

- Since the model requires an efficient text-to-image diffusion model (in terms of number of inference steps), it can hinder the application of  the introduced method with other base models.

**Questions:**

- Since the author roll out full diffusion path (4 steps) during training, how much memory is required for training (e.g., compared to existing methods)?
- How does the model perform if we set the number of diffusion step to (1, 2, 6)? (and how much memory footprint)
- Does this method work for subject other than human as input (e.g., dog, cat, etc) ?

**Limitations:**

please refer to weakness section.

---

> ### Author Rebuttal · Authors · 2024-08-06
>
> Thanks for your positive and valuable comments. Our responses are as follows:
>
> **`Q1:` Since the model requires an efficient text-to-image diffusion model (in terms of number of inference steps), it can hinder the application of the introduced method with other base models.**
>
> Please refer to the detailed explanation in `GQ1`. The fast sampling plugin is not indispensable for our method, which indicates that our method can be adapted to other base models without an acceleration plugin.
>
> ---
>
> **`Q2:` How much memory is required for training?**
>
> We provide comprehensive data in `GQ1` and `GQ2` and discuss the memory issue in the Limitations section of the paper (L464-L465). As most comparative methods did not make their training codes open-source or disclose their memory usage, we cannot make a direct comparison.
>
> ---
>
> **`Q3:` How does the model perform if we set the number of diffusion step to (1, 2, 6)? (and how much memory footprint)**
>
> Please see `GQ2` for a detailed comparison. Note that due to memory constraints, we only update the gradients at 6 of the 8 steps.
>
> ---
>
>
> **`Q4:` Does this method work for subject other than human as input (e.g., dog, cat, etc)?**
>
> Yes, it is theoretically feasible. Please refer to the discussion in `GQ3`.

---

> ### Author Response · Authors · 2024-08-14
> **Further discussions with Reviewer bfEK**
>
> Dear Reviewer bfEK,
>
> We thank you for the precious review time and valuable comments. We have provided corresponding responses, which we believe have covered your concerns. We hope to further discuss with you whether or not your concerns have been addressed. Please let us know if you still have any unclear parts of our work. Thanks :-)
>
> Best,
>
> PuLID Authors

---

### Author Rebuttal · Authors · 2024-08-06

# Global Response

We sincerely appreciate all reviewers for their insightful and valuable feedback. We are delighted that the reviewers find the paper well-written, the method compellingly motivated, the idea innovative, and the experimental results superior and impressive. Below, we address their common concerns.

**`GQ1:` Is the method proposed in the paper heavily reliant on fast sampling techniques? If so, does this imply that the method lacks generalizability, such as being unable to adapt to a model without corresponding fast sampling methods?**

Great question! We want to clarify that the core essence of our paper is to introduce a more accurate ID loss and alignment loss in the T2I training branch to achieve better ID fidelity and editability. Lightning (the fast sampling method) serves as an optional acceleration trick, but it is not indispensable.

Without the fast sampling method, we'd need 30 inference steps with CFG on the T2I branch, compared to the current 4 necessary inference steps without CFG. Due to CUDA memory bottleneck (we exclude the use of gradient checkpointing due to its significant speed penalty), it's not possible to perform backpropagation (BP) of the gradient at all timesteps. However, it remains possible to make optimization viable with strategic techniques. Particularly, for the optimization of ID loss, BP of the gradient happens only for the last few timesteps. For the optimization of alignment loss, a timestep is randomly selected for BP of the gradient. A comparison table below shows the differences in speed and memory consumption between training with and without acceleration.

|     | BP last 1 timestep | BP last 2 timestep | BP last 3 timestep | BP last 4 timestep | BP last 20 timestep |
| -------- | ------- | ------- | ------- | ------- | ------- |
| w/ (4 steps) fast sampling | 2.6s/iter(41GB)    | 2.9s/iter(49GB)    | 3.1s/iter(56GB)    | 3.3s/iter(63GB)    | -    |
| w/o fast sampling   | 6.6s/iter(50GB)    | 7.0s/iter(65GB)    | 7.3s/iter(80GB)    | OOM    | OOM    |

From the table above, we see that if we do not use fast sampling and take SDXL-base as the base model for the T2I training branch, efficiency will indeed be much lower. However, thanks to the carefully designed optimization strategies mentioned above, the training method presented in this paper can be effectively adapted to non-accelerated models, with performance being further improved, shown in the next table.

|               | \|-------- | DivID-120 | --------\| | \|--------  | Unsplash-50 | --------\| |
| ------------- | --------- | --------- | --------- | --------- | -------- | -------- |
|               | Face Sim. | CLIP-T    | CLIP-I    | Face Sim. | CLIP-T   | CLIP-I   |
| w/ (4 steps) fast sampling  | 0.733     | 31.31     | 0.812     | 0.659     | 32.16    | 0.840    |
| w/o fast sampling | 0.743     | 31.75     | 0.842     | 0.687     | 32.58    | 0.865    |

Additional visual results can be found in **Figure 1 of the Rebuttal PDF**. In conclusion, our method does not rely on accelerated base models, thus reflecting the universality of our approach. We also have plans to adapt PuLID to stronger base models and make them open-source in the future, such as FLUX.1, which was recently released.

---

**`GQ2:` Have the authors explored alternative fast sampling methods beyond SDXL-Lightning? Additionally, how effective is the method trained with a differing number of steps (e.g., 1, 2, 8)?**

Firstly, our selection criteria for a fast sampling method is that it can generate high-quality images within a limited number of steps while well preserving the style and layout of the original model. SDXL-Lightning fulfills these requirements and was the SOTA at that time, so we chose it to speed up our T2I training branch. Secondly, we select 4 steps as it balances efficiency and quality. With fewer steps (e.g., 1 or 2), the likelihood that SDXL-Lightning generates flawed faces increases. However, we have noted a recently emerged fast sampling method, Hyper-SD, which performs better than SDXL-Lightning on 1 and 2 steps. Therefore, we present the results of training on 1 step and 2 steps with Hyper-SD, as well as on 8 steps with SDXL-Lightning in the subsequent table.

|               |        |           | \|-------- | DivID-120 | --------\| | \|--------  | Unsplash-50 | --------\| |
| ------------- | ------ | --------- | --------- | --------- | --------- | --------- | -------- | -------- |
|               | Memory | Speed     | Face Sim. | CLIP-T    | CLIP-I    | Face Sim. | CLIP-T   | CLIP-I   |
| Hyper-SD T=1  | 41GB   | 2.2s/iter | 0.694     | 31.91     | 0.819     | 0.632     | 31.89    | 0.857    |
| Hyper-SD T=2  | 49GB   | 2.5s/iter | 0.720     | 32.08     | 0.810     | 0.653     | 32.35    | 0.840    |
| Lightning T=4 | 63GB   | 3.3s/iter | 0.733     | 31.31     | 0.812     | 0.659     | 32.16    | 0.840    |
| Lightning T=8 | 77GB   | 4.1s/iter | 0.734     | 31.66     | 0.818     | 0.668     | 32.19    | 0.850    |

As shown in this table, training with 1 or 2 steps leads to a reduction in face similarity. Meanwhile, training with 8 steps slightly enhances overall performance. Considering both efficiency and performance, 4 steps is a sound choice.


---

**`GQ3:` Can the training method be extended to IP customization task (e.g., clothing, dog, cat)**

Yes, since both IP and ID can be regarded as subjects to some extent, our training method can be adapted to the IP customization task with minor changes. Specifically, the ID loss can be replaced by an objective that measures the similarity between IPs, like CLIP image similarity. The alignment loss does not need to change, but the list of training prompts should be customized for the specific task. Our preliminary experiments indicate that our training method can effectively enhance the prompt-following abilities for IP customization tasks. We will leave this to be explored in future work.

---

### Decision · Program_Chairs · 2024-09-25

**Decision:**

Accept (poster)

**Comment:**

This paper studies tuning-free ID customization for T2I generation. Reviewers gave accept, accept, strong accept, and accept ratings. The ratings are overwhelmingly positive. Reviewers are happy with the good presentation, novelty, performance, and many experiments. The AC recommends accepting this paper as a spotlight presentation.